# Are "Obstetrically Underserved Areas" really underserved? Role of a government support program in the context of changing landscape of maternal service utilization in South Korea: A sequential mixed method approach

Hwa-Young Lee[1], Nan-Hee Yoon[2], Juhwan Oh[3,4]*, Joong Shin Park[5], Jong-Koo Lee[6,7], J. Robin Moon[8], S. V. Subramanian[4]

1 Department of Global Health and Population, Takemi program in International Health, Harvard T.H. Chan School of Public Health, Boston, MA, United States of America, 2 Department of Health Administration, Hanyang Cyber University, Seoul, Republic of Korea, 3 Department of Medicine of Seoul National University College of Medicine, Seoul, Republic of Korea, 4 Department of Social and Behavioral Science, Harvard T.H. Chan School of Public Health, Boston, MA, United States of America, 5 Department of Obstetrics and Gynecology, Seoul National University College of Medicine, Seoul, Republic of Korea, 6 Center for Healthy Society and Education, College of Medicine, Seoul National University, Seoul, Republic of Korea, 7 Department of Family Medicine, Seoul National University College of Medicine, Seoul, Republic of Korea, 8 Bronx Partners for Health Communities New York City, Bronx, NY, United States of America

☯ These authors contributed equally to this work.
* oh328@snu.ac.kr, juhwan.oh328@gmail.com

**Data Availability Statement:** National Health Insurance claims big data that our analyses are

## Abstract

### Objectives

The Korean government has been providing financial support to open and operate the maternal hospital in Obstetrically Underserved Areas (OUAs) since 2011. Our study aims to assess the effectiveness of the government-support program for OUAs and to suggest future directions for it.

### Methods

We performed sequential-mixed method approach. Descriptive analyses and multi-level logistic regression were performed based on the 2015 Korean National Health Insurance claim data. Data for the qualitative analysis were obtained from in-depth interviews with health providers and mothers in OUAs.

### Results

Descriptive analyses indicated that the share of babies born in the hospitals located in the area among total babies ever born from mothers residing in the area (Delivery concentration Index: DCI) was lower in government-supported OUAs than other areas. Qualitative analyses revealed that physical distance is no longer a barrier in current OUAs. Mothers travel to neighboring big cities to seek elective preferences only available at specialized maternal

based on is managed by National Health Insurance Service (NHIS). The database for the purpose of statistical analyses can be accessed by only authorized researchers through remote access system or big data center. The remote access system is to support for analysis through virtualized PC at the user's desired location, and a total of 180 accounts are in operation. The Big data center is a space for data analysis for the research purpose where there are 40 seats in headquarters and branches of NHIS. Interest parties can contact NHIS to obtain the data in the following ways: Tel : 033) 736-2431, 2432, 2433 (information analysis department in big data operation room) Fax: fax033) 749-6337, Web : https://nhiss.nhis.or.kr/bd/ab/bdaba013eng.do.

**Funding:** This paper is based on a research project on "geographical maldistribution of health service in Korea" funded by Division of Public Healthcare, Ministry of Health and Welfare.

**Competing interests:** NO authors have competing interests.

hospitals rather than true medical need. Increasing one-child families changed the mother's perception of pregnancy and childbirth, making them willing to pay for more expensive services. Concern about an emergency for mothers or infants, especially of high-risk mothers was also an important factor to make mothers avoid local government-supported hospitals. Adjusted multi-level logistic regression indicated that DCIs of government-supported OUAs were higher than the ones of their counterpart areas.

## Conclusion

Our results suggest that current OUAs do not reflect reality. Identification of true OUAs where physical distance is a real barrier to the use of obstetric service and focused investment on them is necessary. In addition, more sophisticated performance indicator other than DCI needs to be developed.

## Introduction

High-income countries (HICs) are virtually realizing antenatal care (ANC), birth with skilled birth attendants, and institutional births for all. However, a closer look reveals gaps in care. Quality of maternal care services varies greatly and human resource shortage in remote areas is common, both of which put disadvantaged subpopulations at an elevated risk, causing an inequitable distribution of maternal and child health outcomes [1, 2]. Failure to access maternity service at the right time will cause a delay in the management of obstetric emergencies at delivery, which is one of the main causes of maternal and perinatal morbidity and mortality [3]. Several studies conducted in HICs have reported a positive association between travel time to the maternity ward and risks of intrapartum and neonatal mortality and morbidity [4–8].

The Republic of Korea is notorious for its low fertility rate, recording 1.05 total fertility rate in 2017[9]. This, combined with several other factors such as low service fee for delivery and subsequent low supply of obstetricians, has been driving many maternal hospitals in rural areas to go out of business due to low profitability, contributing to the regional disparity in access to maternal services [10]. Thus, to tackle this problem, Korean government enacted a financial support program to establish a maternal hospital in "Obstetrically Underserved Area (OUA)" in 2011.

### Financial support program for Obstetrically Underserved Areas

To identify the target areas for prioritized support, the government designated the OUAs (where a unit of "area" equals district) based on two criteria: 1) where the proportion of mothers who have delivered their babies at the maternal hospital that can be reached within 60 minutes is less than 30%; or 2) where the proportion of fertile women who cannot reach a maternal hospital within 60 minutes is more than 30% [11]. Distance and time were estimated using Geographic Information System (GIS). Public transportation was the basis of the travel time calculation.

The government started financial support in 2011 to the hospitals that volunteered to open obstetrics services in OUA (hereafter government-supported hospital), which resulted in a total of 35 hospitals or clinics (mostly hospitals) operating through this support program as of the end of 2016. However, most of the mothers were still making a journey to the big city in the neighboring district to utilize better maternal hospitals, especially specialized maternal

hospitals (i.e., accredited by the Ministry of Health (MOH) to provide high-level medical service), rather than seeking government-supported hospital located nearer to them. This raised concerns about the effectiveness of the OUA support program, weakening the justification of expansion of government investment to newly emerging OUAs.

Four rounds of monitoring on the program have been performed and reported notable improvements in some of the maternal and child health indicators and high patient satisfaction with government-supported hospitals, but they reported that the utilization of government-supported hospitals for delivery is still very low [12–15].

However, previous evaluations were based on either qualitative interview results only or simple comparisons of descriptive statistics between areas without consideration of various determinants in the mothers' choice of their healthcare providers. Thus, the present study was conducted with three aims: (i) to identify encouraging and discouraging factors towards the utilization of local government-supported maternal hospitals; (ii) to re-evaluate the effectiveness of the program from a new perspective; and (iii) to suggest future directions for the government-support program for OUAs.

## Study data and methods

### Study design

We employed a sequential mixed-method design integrating the quantitative and qualitative approaches. We initiated descriptive statistical analyses to get a broad picture of the location of the mother's obstetric service utilization (phase 1). A subsequent qualitative interview was designed to identify facilitators and barriers to the utilization of government-supported hospitals based on phase 1 results (phase 2). The results from phase 2 qualitative study were again incorporated to guide the design of the main quantitative analyses at the final phase (phase 3) [16].

### Study 1: Quantitative component (phases 1 and 3)

**Data.**   For quantitative analyses, medical records for delivery and antenatal care that occurred from January to December in 2015 were extracted from the Korean National Health Insurance (NHIS) claims database. This database contained claims data for all services provided through the National Health Insurance (NHI) program and the Medical Aid program. The NHI program of Korea covers 96~97% of the population as compulsory social insurance. Due to the nature of the data, a unit of analysis is a medical event such as delivery or use of antenatal care service. Information on district-level variables was derived from an administrative database provided by the Korean Statistical Information Service (KOSIS) [17] and Ministry of Interior and Safety [18].

**Variables.**   For descriptive and main analyses, we operationally defined "delivery concentration index (DCI)" as a share of infants that were born at local hospitals located within the residential area among the infants ever born to mothers living within the area regardless of geographical location of delivery. We grouped the entire areas into five categories based on DCI and OUAs defined by a government (Fig 1)

First, the "delivery-concentrated area (DCA)" is where DCI is more than one, indicating an influx of pregnant mothers from other areas for delivery. Areas with less than one of DCI were categorized into three: potential OUAs, OUAs, and others. "Potential OUAs" indicates where maternal hospitals in operation exist but with high concern for closure due to ever-declining number of childbearing age women in the area. OUAs divided again into two; "government-supported" and "non-government supported". Finally, "others" includes all remaining areas that belong to none of the categories above (Fig 1).

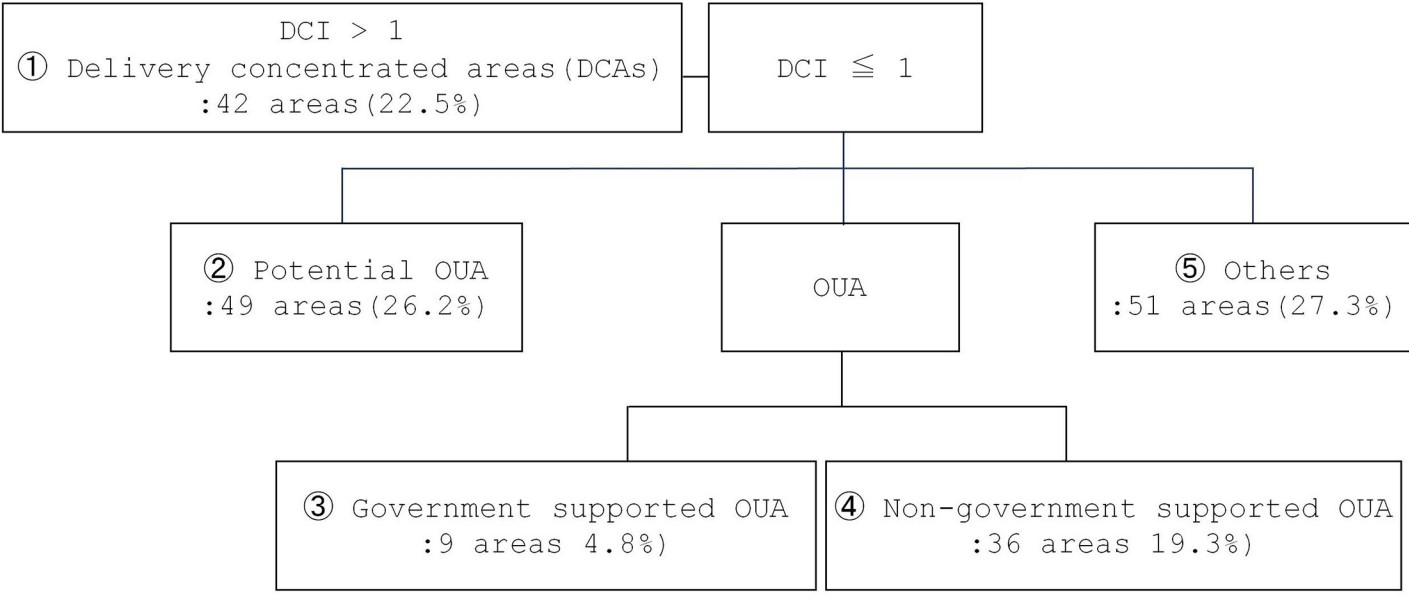

**Fig 1. Classification of areas.**

Two outcome measures were employed for main analyses: whether mothers utilized maternal hospital located in their residential area for delivery which was operationalized as a binary response and the proportion of the number of antenatal care visits to the hospital located in their residential area to the total number of antenatal care visits, which ranges from 0 and 1 and thus, was operationalized as a continuous variable.

The main interest independent variable is the type of residential area where mothers reside (government-supported OUAs vs. comparison areas (potential OUA (②) or others(⑤)). We excluded DCAs from the analyses because the government-support program did not aim to improve DCI in OUAs to the level in DCAs and therefore, comparison with DCAs would be meaningless. Non-government supported OUAs were additionally excluded because those areas do not have any local maternal hospitals, meaning that DCI is inevitably zero.

Other covariates were chosen by reviewing previous articles and the results from phase 2 qualitative interviews: Maternal age (≧35 vs. <35 years old), income level (proxied by health insurance premium, which is proportional to income or asset: enrollees in Medicaid, quartiles of the health insurance premium for health insurance enrollees), job status (employed vs. unemployed), high-risk status (yes vs. no), delivery type (natural delivery vs. Caesarian section), and primiparity (primipara vs. multiparous). The high-risk group was defined as mothers with a history of preterm labor, preterm delivery, hemorrhage related to delivery, pre-eclampsia or eclampsia, multiple birth, or breech delivery. District-level variables included a total number of births, financial self-reliance ratio, and existence of postnatal care centers.

Phase 1 descriptive analyses include the distribution of deliveries by district, comparison of DCI between government-supported OUAs and comparison areas, and finally, the comparison of study sample characteristics of the government-supported OUAs and comparison areas.

For main analyses at phase 3, two-level random intercept logistic (for the outcome variable indicating whether mothers utilized maternal hospital located in their residential area for delivery) and linear regression (for the outcome variable indicating the proportion of the number of antenatal care visits to the hospital located in their residential area to the total number of antenatal care visits) was employed with medical events at level-1 nested within districts at level 2 to assess

the effect of the government-support program on mothers' choices of a maternal hospital when other relevant factors were adjusted for. Model 1 only included terms representing individual mothers' characteristics while model 2 additionally included district characteristics to model 1.

### Study 2: Qualitative component

**Data.** In-depth interviews were conducted in person by three authors alternately in pairs using a semi-structured questionnaire drawn up based on results from descriptive analyses and previous articles. Original questionnaires both in Korean and translated in English are included in a supplemental file to this manuscript (S1 File). Data were collected between November and December in 2016.

**Ethical statement.** The study protocol was approved by the Research Ethics Committee at Seoul National University Hospital. All interviewees were given study information and provided written consent to participate.

**Sample.** Our interview sample composed of three groups: 1) mothers who did and did not use local maternal hospitals supported by the government were recruited using a purposive sampling strategy and interviewed to identify facilitators and barriers to the utilization of government-supported hospitals, named "demand-side factors". Purposive sampling strategy is used when researchers assume that certain categories of individuals may have a unique, or important perspective on the phenomenon in question [19]. Since the authors did not have direct access to the patient information, we asked the government-supported hospitals to recruit mothers who recently had delivered a baby in their hospital. To recruit the mothers who had delivered outside their residential areas, we contacted and asked personnel in charge of maternal and child health in local health centers because they have almost every information on the pregnancies and births that had occurred within their territory. 2) Doctors, nurses and hospital managers in government-supported hospitals were interviewed to identify the difficulties in running the OB/GY services based on government funding, named "supply-side factors". 3) Finally, personnel in charge of maternal and child health in local health centers in OUAs were additionally interviewed to examine the difficulties in implementing the government-support program. Doctors, nurses, and hospital managers and personnel in local health centers were recruited by convenience sampling. Convenience sampling might have a danger that generalization is unwarranted when the sample universe is broad. However, this concern would be insignificant because our interview covered 18 of 35 government-supported hospitals, which amount to more than 50% of the entire sample.

**Analysis.** All the interviews were tape-recorded and transcribed. To maximize the rigor, three authors who conducted interviews independently read and coded the randomly allocated batches of interview transcripts using grounded theory methodology [20] and reached agreement on the emerging themes and subthemes through discussion. Specifically, the constant comparative method was employed to deduct barriers and promoters to the utilization of local government-supported maternal hospital in their residential area and difficulties in operating the support program. Each word and sentence were analyzed to identify tentative themes, which were compared with each other within and across transcripts. Themes with similar meanings were grouped together and finally, categories were formulated from the themes. We dropped the themes that were not within the scope of our immediate interest.

## Results

### Phase 1: Descriptive statistics

There was an apparent unequal distribution in the use of delivery service between districts. Fig 2 shows a distribution of deliveries with the X-axis representing a cumulative share of districts

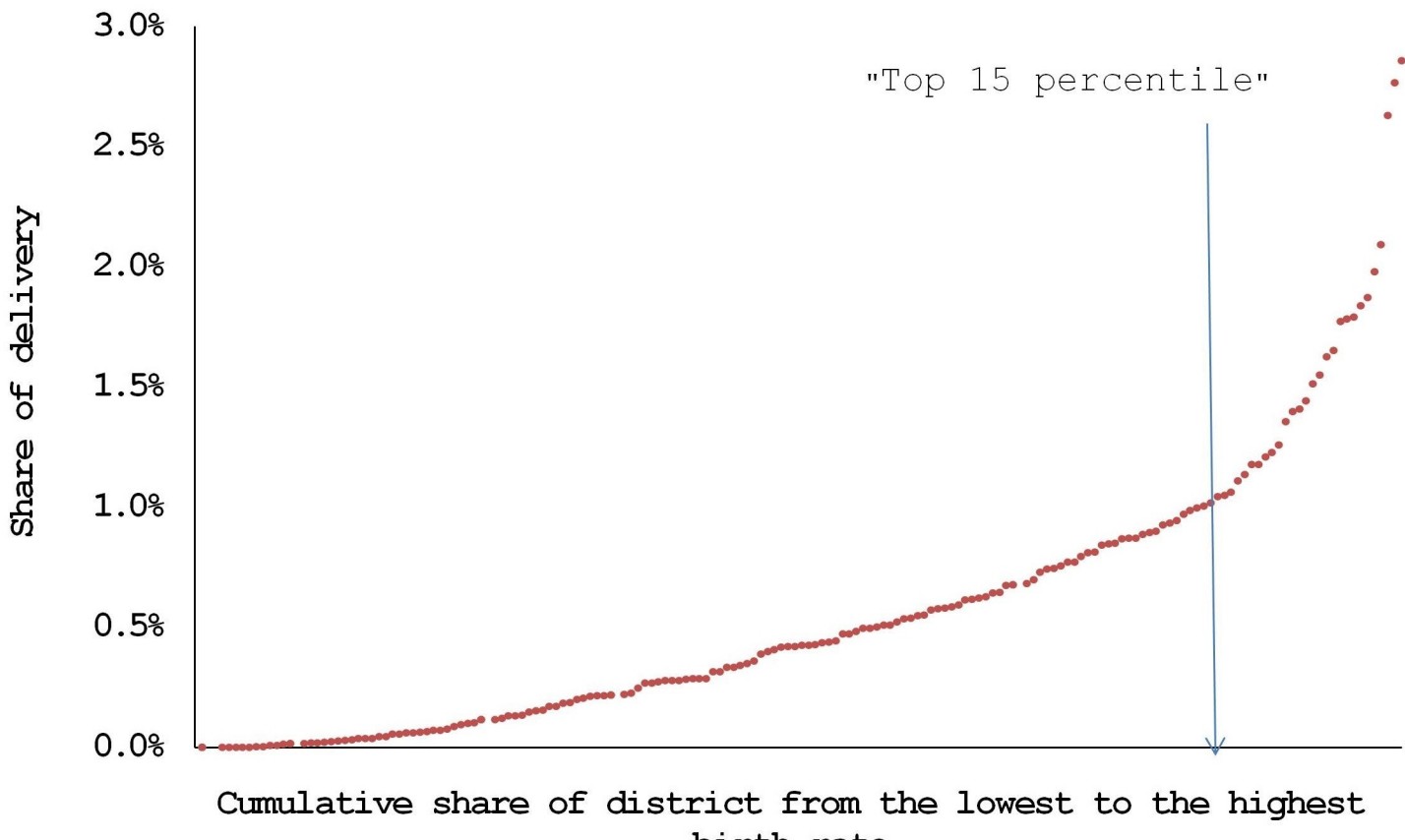

**Fig 2. Distribution of the deliveries by the district.**

from the lowest birth rate to the highest and the Y-axis plotting share of the number of deliveries in the relevant district among the total number of deliveries of the Republic of Korea in 2015. The share of deliveries in the district sharply increases at the point of approximately the top 15 percentile. Specifically, the number of deliveries that had happened in 71 districts belonging to the DCA category accounted for about 73.3% of the total delivery that happened across the country (393,826 deliveries in 266 districts in total), meaning that three-fourths of the deliveries happened in only one-fourth of the districts. DCIs in top 10 districts were higher than three, which means two-thirds of babies born in the top 10 districts were from mothers residing in other districts.

After excluding DCAs and non-government funded OUA, we compared the DCIs among government-support OUAs, potential OUAs, and "others." The grand mean of DCI of all the 266 districts was 0.69. DCI in government-support OUAs was 0.22 on average, higher than in potential OUA (0.12), but still lower than "others" (0.34) as shown in Fig 3.

Characteristics of government-supported OUAs and comparison areas composed of potential OUAs and "others" were compared (Table 1). After excluding the DCAs, the total number of deliveries was 120,472, of which 3,050 deliveries occurred in government-supported OUAs. Since the unit of observation is a delivery event, mothers who gave births twice in 2015, albeit negligible, were counted separately, which is logically valid because even same mother may make a different decision in choosing a hospital at each time of delivery.

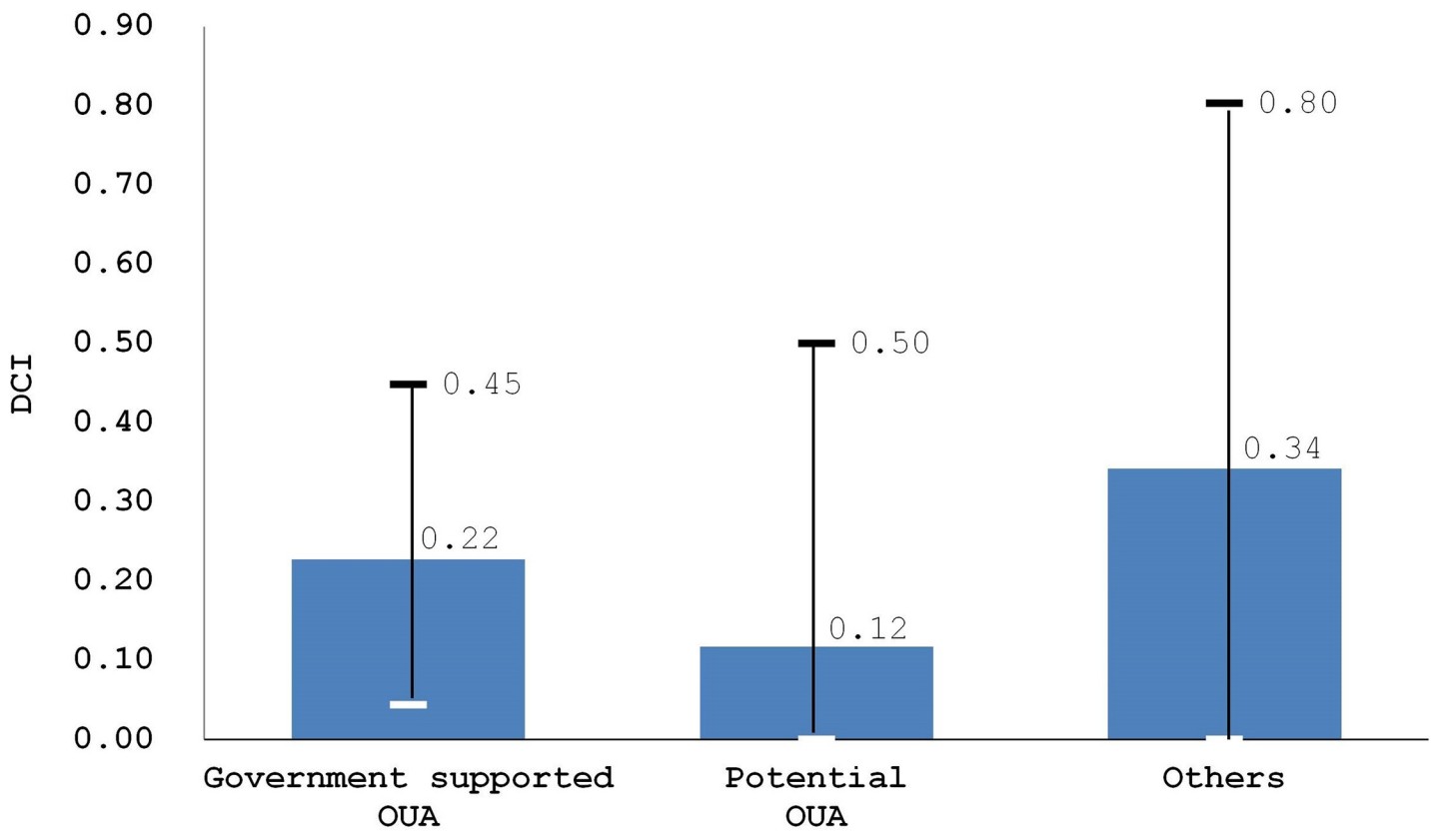

**Fig 3. DCI by district types.**

**Table 1. Characteristics of the study sample by the type of residential area.**

| Variables | Categories | Government- supported OUAs | | Comparison areas | | X² (p-value) |
|---|---|---|---|---|---|---|
| Age (years) | <35 | 2,325 | (76.2) | 89,821 | (76.5) | 0.116 |
| | 35 ≤ | 725 | (23.8) | 27,601 | (23.5) | (0.73) |
| Job | Unemployed | 2,127 | (69.7) | 81,304 | (69.2) | 0.345 |
| | Employed | 923 | (30.3) | 36,118 | (30.8) | (0.56) |
| Wealth quartile | Poorest (= Medicaid) | 219 | (7.2) | 7,124 | (6.1) | 105.172 |
| | 1st | 457 | (15.0) | 13,960 | (11.9) | (< .001) |
| | 2nd | 761 | (25.0) | 24,928 | (21.2) | |
| | 3rd | 1,146 | (37.6) | 46,127 | (39.3) | |
| | 4th (wealthiest) | 467 | (15.3) | 25,283 | (21.5) | |
| High risk | No | 2,545 | (83.4) | 97,742 | (83.2) | 0.088 |
| | Yes | 505 | (16.6) | 19,680 | (16.8) | (0.77) |
| Delivery method | Natural delivery | 1,830 | (60.0) | 68,896 | (58.7) | 2.157 |
| | Cesarean section | 1,220 | (40.0) | 48,526 | (41.3) | (0.14) |
| Parity | Primiparous | 1,466 | (48.1) | 60,464 | (51.5) | 13.979 |
| | Multiparous | 695 | (51.9) | 56,958 | (48.5) | (< .001) |
| Delivery area | Within residence | 2,355 | (22.8) | 33,561 | (28.6) | 49.053 |
| | Outside residence | 2,127 | (77.2) | 83,861 | (71.4) | (< .001) |
| Total | | 3,050 | (2.5) | 117,422 | (97.5) | |

Mother's age, job status at delivery, high-risk status, and type of delivery were not statistically significantly different between the two groups. On the other hand, mothers who were residing in government-supported OUAs were poorer, higher in the proportion of primiparity, and also higher in the proportion of delivery outside their residence with statistical significance.

## Phase 2: Qualitative study

Total of 109 interviewees participated in interviews. The characteristics of interviewees were presented in S1 Table.

**Facilitators to the use of government-supported hospital.** *Proximity.* Proximity was the strongest motive for using the government-supported maternal hospital. Despite the improved road condition connecting district to district and increased personal vehicle ownership, some mothers, especially those who cannot drive on their own or who have another child too young to travel with, still find it difficult to travel back and forth almost one-hour distance. Also, family members and relatives can easily visit to see mother and baby in a nearby hospital during the post-partum period.

*"I already had a first child who was one-year-old when I got pregnant (second baby). I delivered the first child at a hospital in An-Dong (a big city located in the neighboring district). However, I couldn't go that far during the second pregnancy because there was nobody to care for my first baby. So I gave birth to my second baby in this (government-supported) hospital. I did not experience any inconvenience during my stay. The most satisfying factor was that hospital is so close to my home that my husband and first baby could easily visit (YMOKB1)."*

**Quality of care.** Quality of care was the next important reason for the choice of government-supported maternal hospital. Duration of doctor consultation is very short in most of the specialized maternal hospitals in a big city (mostly DCA). Most of those hospitals are large in scale and thus, input cost is very high which motivates them to see as many patients as possible in order to maximize profit. Therefore, patients cannot have quality and sufficient time with doctors during their consultation and do not feel comfortable asking questions to doctors.

*"An-dong hospital (specialized maternal hospital) was generally good. However, wait time was so long and consultation was not satisfying. I could not find a chance to ask a question because doctors always looked busy and in a hurry (NOMOKB3)."*

*"During the delivery of my second child (at the specialized maternal hospital in Kwang-Ju), they were unkind and their attitude towards patients was mechanical. I even saw them (formula) feeding a baby without holding, but just laying them on the cradle (YMOJN4)."*

On the other hand, the government-supported hospitals are generally less profit-seeking. They had voluntarily participated in the government-support program in full knowledge of the circumstance that government subsidy is far from sufficient so they could be at a financial loss, much less making a profit. Therefore, doctors in the government-supported hospital afford the time to see patients more thoroughly and kindly than those in specialized maternal hospital. Staff at the government-supported hospitals also give priority to patients' emotional well-being throughout their stay at the hospital.

*"They do sonogram very carefully. They did not hurry even though the waiting line is long. Sometimes they spend about 30 or 40 minutes on only one patient (YMOJN2)."*

**Low cost.**  Local government in OUAs provide additional financial support for pregnant mothers such as a voucher for an antenatal check-up or cash subsidy for delivery when mothers use the local government-supported hospital in an effort to raise fertility rate in their district. Such an incentive to mothers is not provided when they use hospitals outside their residential district. The specialized maternal hospitals, normally located in DCA, tend to induce demand for non-essential services which are not covered by health insurance. This drives service cost even higher.

> *"There was no extra charge for a single room or meal (at the government-supported hospital) while I had to pay for them during my last delivery (at the specialized maternal hospital). I think. . . total cost (at the government-supported hospital) was roughly three-fifths of the cost I paid during my last delivery (YMOKB1)."*

Next, we identified a few deterrent factors which led mothers to choose to travel to the city in the neighboring district for the bigger and specialized maternal hospital over the government-supported hospital. The comparative method produced several detailed sub-themes, which were clustered into two broad themes again: demand-side and supply-side factors.

**Demand-side barriers to the use of government-supported hospital.**  *Concern about an emergency.* Mothers have great concerns about potential emergencies that might occur to themselves or infants during delivery. Proper neonatal care, a key for perinatal survival, is not available in most of the government–supported hospitals. For example, some hospitals do not have pediatric department. Even hospitals having a pediatric department rarely have doctors specifically trained for infant care nor are they properly equipped with medical appliances for an emergency. Although government–supported hospital has a referral system to a higher level facility in case of an infant or maternal emergency, mothers still worry about the trouble that may occur during transfer.

> *"I had a fear about a delayed response in case of an emergency that might occur to my baby. While newborn babies can fall into an emergency very quickly, they (government-supported hospital) are not properly staffed or equipped for it (NOMOKB2)."*

Sometimes, mothers choose a bigger hospital in a neighboring city not by their own preference but by their doctors' decision. Doctors in government-supported hospital transfer mothers if there is any possibility of premature delivery, long before due date from fear of unexpected medical accidents.

> *"We refer high-risk mothers to the hospital located in Dae-Gu after discussing with a pediatrician. We don't deal with babies born at less than 34 weeks' gestation. Lungs of those preterm babies of less than 34 weeks are not fully developed, We don't have proper equipment to deal with it (DRKB1)."*

Pregnant mothers with high-risk factors such as mother's age older than 35, pre-eclampsia or eclampsia, and gestational diabetes are also being transferred by doctors to higher-level facilities located in a big city. Overall, doctors in government-supported hospital tend to be risk-averse in serving patients.

**Lack of a post-partum center in the area.**  Korea has a unique post-partum culture that most of the mothers stay in a post-partum care center for a few weeks after giving birth where mothers are provided with all-inclusive services 24 hours a day, various programs for

recuperating themselves to the normal condition such as special meals, and massages, and learning how to care new-born babies. While a majority of the post-partum centers are private and expensive, the use of the post-partum center is culturally considered as a "must-do" rather than "nice to do" to pregnant mothers. However, there is no post-partum center in OUAs because the low fertility rate in OUAs makes the post-partum center business unprofitable. Because long-distance travel is not optimal immediately after giving birth, many pregnant mothers who were willing to use government-supported hospitals end up choosing a bigger specialized maternal hospital with a post-partum center nearby.

> *"I used a nearby hospital (government-supported hospital) when I delivered my first child. I hired a full-time caregiver at home then. However, I still had to do lots of household chores during the night after she left in the evening. So I delivered my second baby in the specialized maternal hospital located in a neighboring district and stayed at post-partum center after delivery. I could focus only on restoring myself there (NoMoCB2)."*

**Improved transportation and road network.** Improved road condition and travel environment has removed physical barriers to travel to a big city of the neighboring districts. Besides, while the definition of OUA was based on travel time by public transportation, it was found that almost every household, rich or poor, owns a private vehicle.

> *"It takes only about 40–50 minutes (to go to the hospital in a neighboring district). In my first delivery, my water (amniotic fluid) broke at home. But, I went to the hospital (in a neighboring district) and delivered safely without any problem (MNOCB3)."*

> *"Car is essential for living in this remote area. A big city or downtown has a good public transportation system but here is different. We cannot go anywhere unless we own a car (HCKB2)"*

Due to this, they were already recognizing neighboring areas as their own sphere of life. Traveling to a big city in neighboring districts was not a trouble to them, but rather a welcome opportunity for getting out of daily routines not only for a hospital visit but also for shopping or entertaining.

> *"In Young-Dong (OUA), there is no entertainment or recreational facilities such as theater or big shopping place. Our family usually go to the neighboring city on weekends for hospital check-up as well as for shopping and dining out (UMOCB3)."*

**Preference for a specialized maternal hospital over a general hospital.** Recently, specialized maternal hospitals are getting popularity among young mothers. They prefer it to general hospitals due to a few reasons. First, specialized maternal hospitals are equipped with high tech diagnostics such as 4D sonograms and provide various options for delivery such as Lamaze or Leboyer. Second, they can have the benefit of economies of scale, being able to afford to have sufficient human resources such as doctors, frontline staff and physical spaces. This enables them to operate longer hours, even opening on weekends and to arrange their space in a way that outpatient clinics, wards, and the ICU are separated.

> *"XX hospital (government-supported maternal hospital) has a delivery room, newborn unit, and ICU located next to each other. In addition, pediatrics outpatient clinics are right next to the OB/GY inpatient wards, which can expose mothers who just delivered a baby to infection from outpatient (NOMOCB2)."*

**Change in attitude and culture on pregnancy and childbirth.**   Crowding into specialized maternal hospitals and increased use of post-partum center after delivery despite the high cost reflect a change in attitude and culture on pregnancy and childbirth. As one-child families are increasing, the mother's perception toward pregnancy and childbirth is changing from just common experiences to a once-in-a-lifetime event. Therefore, patients and families invest time and money during pregnancy and delivery more than ever, seeking high quality and state-of-art service, and sometimes spending more than they can afford. A substantial proportion of mothers who have used the government-supported hospital in OUAs were from poor households who could not afford these kinds of services.

*"Young mothers want to enjoy as much as they can during their pregnancy and delivery, the same context as we want to eat out at luxurious and expensive dining places on our wedding anniversary because it is just rare event in life (NRJN1)."*

Specialized maternal hospitals have emerged along with this new culture, providing various special services during antenatal, delivery and the post-partum period, which are not medically necessary but very attractive electives to mothers.

*"I had much interest in Leboyer delivery. But, I had to give up because the nearby hospital (government-supported hospital) could not do that. They don't even have family delivery room (NOMOKN3)."*

**Supply-side barriers to the success of the government-support program.**   We identified several difficulties that government-supported hospitals have experienced, which make hospitals in other OUAs hesitate to volunteer for participation in the government-support program.

**Difficulty in recruiting health professionals.**   Most of the government-supported hospitals were struggling to recruit medical staffs, especially pediatricians, anesthesiologists, and nurses. Healthcare professionals avoid working in OUAs due to suboptimal working conditions, unsafe working environments, unpromising educational and career development opportunities, and limited educational opportunities for their children.

*"We are having huge difficulties recruiting pediatrician, especially specialized in neonate. So we have to refer premature babies to bigger hospitals in the neighborhood district. However, even they don't want to take premature baby patients (HC1KW)."*

**Insufficient fund.**   he current subsidy can barely afford to cover payrolls for a minimum number of personnel required to get a government subsidy (two doctors and seven nurses). Because the revenue from patients in OB/GYN department is not enough to cover payment for other operating costs to provide the maternal service, government-supported hospitals were compensating the loss in the OB/GYN department by the profit from other departments. Most of the government-supported hospitals did not have a financial fallback mechanism prepared for a medical accident.

*"Medical malpractice is more frequent in OB/GY department than other ones. Doctors have to bear the cost for their malpractice because we don't have any insurance and the like. So we see patients in a very defensive way (DRKB2)"*

**Inadequacy of a performance indicator of the support program.**   Although there is no official indicator to measure the performance of the support program, MOH is implicitly

using DCI to monitor the progress of the support program, and the still low DCI in OUAs they are supporting makes them hesitate to raise the grant money or expand the support to emerging OUAs. However, most of the government-supported hospitals shared the opinion that DCI-based evaluation is unreasonable because there are various factors other than just proximity to the hospital in a mother's hospital choice. In addition, the government-supported hospitals believe that they are contributing in terms of provision of not only delivery service but also gynecological services for the elderly which can lead to early detection of potential severe diseases. However, these are not acknowledged as contributions by the government evaluation.

> *"We think the value of this government-support program is that we can care for the mothers who cannot go far to hospitals due to various reasons no matter how small they may be in number. It is a government's responsibility to make an environment where every pregnant woman can utilize the maternal service without geographical limitation. If the government stop the support just because it could not show a big increase in the number of delivery cases within the district, those women would lose their right for access to basic health service (MOHJN1)"*

## Phase 3: Multi-level logistic regression

Further analyses using a multi-level logistic and linear regression were performed to compare the choice of hospital for their delivery and antenatal check-up between OUAs and comparison areas while other factors elicited from the qualitative interviews as well as review of relevant previous studies were adjusted for.

No statistically significant difference was found in the odds of utilizing the hospitals located in their residential area for delivery (Table 2) as well as in the proportion of the number of ANC visits at the hospitals located in their residential area among the total number of ANC visits (Table 3) between mothers residing in government-supported OUAs and the comparison areas when adjusted for only individual-level characteristics. However, when district characteristics were additionally adjusted for such as a total number of newborn babies within the district, fiscal self-reliance ratio and the existence of post-partum care center in the area, odds of using hospital in their residential area for delivery of mothers residing in government-supported OUAs (that is, government-supported hospital) was almost 12.8 times as high compared to mothers residing in comparison areas, indicating that district characteristics are more contributing than mother's characteristics in the mother's choice of hospital. Percentage of variance attributable to the district was approximately 67.0%, which decreased to 54.3% after adjustment of district characteristics (Table 2). Similarly, the proportion of the number of ANC visits at the hospitals located in their residential area among the total number of ANC visits was 0.159 higher among mothers residing in government-supported OUAs (that is, government-supported hospital) compared to mothers residing in comparison areas after adjusting for district-level characteristics. Adjustment of district characteristics reduced the percentage of variance attributable to the district from 24.9% to 15.0% (Table 3).

All the mother's characteristics were significant predictors of choice of hospital for their delivery and ANC. Mothers who were employed, in higher economic status, had high-risk factors, had Cesarean section, and were in first pregnancy were less likely to use the hospital located within their residential area for delivery compared to their counterparts. On the other hand, mothers who had Cesarean section and were in first pregnancy were more likely to use the hospital within their residential area for their antenatal care.

**Table 2. Results from multi-level logistic regressions on the association between type of residential area and choice of a hospital for delivery.**

| Variables | Model 1 | | | Model 2 | | |
|---|---|---|---|---|---|---|
| | OR | 95% CI | P value | OR | 95% CI | P value |
| **Fixed part** | | | | | | |
| **Individual level** | | | | | | |
| Maternal age (Ref <35) | | | | | | |
| 35≤ | 0.97* | (0.93–1.00 | 0.072 | 0.97 | (0.93–1.00) | 0.071 |
| Job (Ref = non-employed) | | | | | | |
| Employed | 0.79 | (0.76–0.81) | < .0001 | 0.79 | (0.76–0.81) | < .0001 |
| Wealth level (Ref = Medicaid enrollee) | | | | | | |
| 1st quartile§ | 0.84 | (0.78–0.91) | < .0001 | 0.84 | (0.78–0.91) | < .0001 |
| 2nd quartile | 1.00 | (0.95–1.06) | 0.905 | 1.00 | (0.95–1.06) | 0.907 |
| 3rd quartile | 0.84 | (0.80–0.88) | < .0001 | 0.84 | (0.80–0.88) | < .0001 |
| 4th quartile‡ | 0.67 | (0.64–0.71) | < .0001 | 0.67 | (0.64–0.71) | < .0001 |
| High risk (Ref = no) | | | | | | |
| High-risk | 0.59 | (0.56–0.61) | < .0001 | 0.59 | (0.56–0.61) | < .0001 |
| Delivery (Ref = natural delivery) | | | | | | |
| Caesarean section | 0.96 | (0.93–0.99) | 0.015 | 0.96 | (0.93–0.99) | 0.015 |
| Parity (Ref = multiparous) | | | | | | |
| Primiparous | 0.61 | (0.60–0.63) | < .0001 | 0.61 | (0.60–0.63) | < .0001 |
| **District level** | | | | | | |
| Number of newborn babies | | | | 1.00** | (1.00–1.00) | 0.004 |
| Fiscal self-reliance ratio | | | | 0.96 | (0.91–1.01) | 0.102 |
| Post-partum care center (Ref = no) | | | | | | |
| Yes | | | | 18.10 | (5.50–59.55) | < .0001 |
| Area (Ref = comparison areas) | | | | | | |
| Government-supported OUA | 2.54 | (0.42–15.33) | 0.312 | 12.87 | (2.92–56.79) | 0.001 |
| **Random Part** | | | | | | |
| Between-district variance (SE) | 6.68(1.20) p < .0001 | | | 3.90(0.71) p < .0001 | | |
| ICC | 67.0% | | | 54.3% | | |

(§: poorest among health insurance enrollee/‡ richest among health insurance enrollee)

ICC: Intra-class coefficient

## Discussion

Unavailability of or inaccessibility to obstetric care in OUAs has led to the enactment of the government-support program in South Korea. However, there has been an increasing concern that the utilization of newly opened maternal hospital in OUAs supported by the government remains low. The government had concerns that the support program was not as effective as they had expected it to be. Thus this research was performed to evaluate the effectiveness of the program.

Findings from descriptive statistics indicate that DCI in government-supported OUAs was lower compared to the areas belonging to "others", which means pregnant mothers are still traveling to other cities for delivery even though maternal hospitals become available in their residential areas. In-depth interviews with mothers residing in OUAs who did or did not use government-supported hospitals revealed a few factors hampering the utilization of government-supported hospitals. First, proximity to the hospital is no longer an important factor in mothers' choice of hospital. Rather, most of the mothers are willing to travel far to use specialized maternal hospitals which have state-of-the-art equipment or provide various "(so-called)

**Table 3. Results from multi-level linear regressions on the association between type of residential area and choice of a hospital for antenatal care.**

| Variables | Model 1 | | | Model 2 | | |
|---|---|---|---|---|---|---|
| | b | 95% CI | P value | b | 95% CI | P value |
| **Fixed part** | | | | | | |
| **Individual level** | | | | | | |
| Maternal age (Ref <35) | | | | | | |
| 35≤ | 0.018 | (0.013, 0.023) | < .0001 | 0.018 | (0.013, 0.023) | < .0001 |
| Job (Ref = non-employed) | | | | | | |
| Employed | -0.047 | (-0.051, -0.042) | < .0001 | -0.047 | (-0.051, -0.042) | < .0001 |
| Wealth level (Ref = Medicaid enrollee) | | | | | | |
| 1st quartile§ | -0.014 | (-0.025, -0.004) | 0.008 | -0.014 | (-0.025, -0.004) | 0.008 |
| 2nd quartile | -0.003 | (-0.010, 0.003) | 0.326 | -0.003 | (-0.010, 0.003) | 0.323 |
| 3rd quartile | -0.033 | (-0.028, -0.016) | < .0001 | -0.022 | (-0.028, -0.016) | < .0001 |
| 4th quartile‡ | -0.051 | (-0.058, -0.044) | < .0001 | -0.051 | (-0.058, -0.044) | < .0001 |
| High risk (Ref = no) | | | | | | |
| High-risk | -0.044 | (-0.050, -0.039) | < .0001 | -0.044 | (-0.050, -0.039) | < .0001 |
| Delivery (Ref = natural delivery) | | | | | | |
| Caesarean section | 0.248 | (0.138, 0.359) | < .0001 | 0.248 | (0.138, 0.359) | < .0001 |
| Parity (Ref = multiparous) | | | | | | |
| Primiparous | 0.074 | (-0.031, 0.180) | 0.169 | 0.074 | (-0.031, 0.180) | 0.168 |
| **District level** | | | | | | |
| Number of newborn babies | | | | 0.000 | (0.000, 0.000) | < .0001 |
| Fiscal self-reliance ratio | | | | -0.005 | (-0.008, -0.002) | 0.003 |
| Post-partum care center (Ref = no) | | | | | | |
| Yes | | | | 0.227 | (0.145–0.310) | < .0001 |
| Area (Ref = comparison areas) | | | | | | |
| Government-supported OUA | 0.091 | (-0.050, 0.231) | 0.209 | 0.159 | (0.054, 0.264) | 0.004 |
| **Random Part** | | | | | | |
| Between-district variance (SE) | 0.043 (p: <0.001) | | | 0.023 (p: <0.001) | | |
| ICC | 24.9% | | | 15.0% | | |

(§ : poorest among health insurance enrollee /‡ richest among health insurance enrollee)

ICC: Intra-class coefficient

high-quality" maternal services, either due to their concerns about potential emergency or because they want to enjoy more pleasurable (not medically necessary) services during their pregnancy and delivery. This phenomenon is not limited to Korea but occurs in other high-income countries. It was reported that Dutch patients use the information on the quality of care more than on general hospital information including distance to the hospital when they choose the hospital for the surgery [21]. Another study reported that patients in the UK are willing to travel depending on accessibility to services, the organization of the services, socio-economic characteristics of mothers or perceptions about the providers [22, 23]. Although previous evaluations on government-support program have identified reliable quality of hospital facilities and system as one of the main factors for utilizing the hospital outside their district [12, 14], recommending to improve the quality of the government-supported hospital. none of the previous evaluations has not brought the changing preference of mothers for luxurious services or delivery culture among young mothers to their attention.

Second, most of the interviewed mothers put much priority on the availability of post-partum care center nearly. Importance of post-partum care centers in mother's choice of delivery

hospital has grown, given the result of previous evaluation performed in 2014 that only small portion of mothers (less than 6%) cited a lack of post-partum care center as a reason of not using government-supported hospital [12, 24]

Finally, due to improved road connections and thus shortened travel time to neighboring districts, mothers hardly recognize traveling to the big city as a barrier. A similar matter was once pointed out in previous evaluation performed in 2013 but, without placing much emphasis on it. 17% of respondents answered that it took less than 30 minutes to reach the hospital located in a neighboring city [14], about which they recommended that, in addition to the administrative boundary, actual accessibility to the hospitals or population distribution needs to be considered when they choose the location of the government-supported hospital.

There are also a few other difficulties that government-supported hospitals are suffering from such as insufficient funds, shortage of healthcare professionals in the area, and pressure to show improvement measured by the inadequate/incomplete performance indicator (DCI).

Multi-level analyses while adjusting for maternal and area characteristics identified from interviews and review of previous articles, mothers residing in government-supported OUAs are about 12 times more likely to utilize local maternal hospitals in their residential area for their delivery and antenatal care compared to those in the comparison areas. In other words, if OUAs are composed of mothers with the same characteristics and have the same district factors as a comparison area, DCI would be higher in OUAs than in comparison areas.

Our findings offer a few insights into future directions of the government OUA-support program. First, the current definition of OUA, which was based on travel time to reach the nearest maternal hospital by public transportation does not fully reflect reality. It turned out that almost every household, rich or poor, has vehicle ownership and road network connecting districts are rapidly being expanded. Consequently, most women in OUAs can reach the nearest maternal hospital under 60 minutes. A new definition reflecting improved road network and vehicle ownership would identify a smaller number of OUAs than the current list. Second, the performance indicator for support program should be diversified. A simple comparison of unadjusted DCI between the government-supported OUAs and other areas is unfair and disregards reality. For example, post-partum care centers, which proved to be the most influential factor for mothers to choose a hospital for their delivery in the multi-level analysis as well as in the interviews, exist in only one district among OUAs. In addition, mothers prefer high tech equipment and fancy services and amenities and are willing to invest more in pregnancy and delivery because more and more couples are having only one child. In this context, attaining the same level of DCI as other areas with the minimum government fund is not feasible. A simple comparison of DCI without considering other factors can undermine the morale of government-supported hospitals and discouraging hospitals in other OUAs from newly joining in the government-support program. Comparison considering population characteristics and environmental factors of the area is required. Also, improved accessibility to gynecologic outpatient services needs to be recognized as a support program achievement.

Last but not least, the government needs to reconsider the ultimate value of the support program. It was found that most of the mothers do not mind traveling out of their residential areas for hospital visits. Rather, it is a welcome opportunity for them to go out of their daily routines to find a recreational venue for shopping or eating out. Considering this, aiming to induce the mothers, who happily travel to the neighboring district to seek more specialized service to use the hospital in their residential area would be pointless.

There are several limitations to be considered when interpreting the results. First, we used the information on the national health insurance premium as a proxy for economic status because NHIS data does not have the information on the actual income. Korean National Health Insurance premium is based on two contribution scheme: 1) for industrial workers and

government and school employees, the contribution is proportional to wage income 2) for the self-employed, the contribution is calculated with a formula based on income and property because the incomes of the self-employed are only partially captured. Therefore, there might be a mismatch between the insurance premium and actual economic status. However, we assume that error between captured by insurance premium and actual income would be minimal. Also, we operationalized variable as quintile rather than continuous one so chance of miscategorization into the wrong wealth quintile would be low. Another limitation lies in the fact that although interviewing a diverse range of individuals enables triangulation of collected information, and thus allows a better assessment of the generality of the explanation that one develops, we could not confirm that the mother interviewees were from various background such as socioeconomic status because we recruited them indirectly through the government-supported hospitals or public health center staffs. However, we tried to triangulate the interview data as extensively as possible among the mothers, providers, and staffs at health centers.

On the other hand, this study has several strengths that distinguish the present study from the previous evaluation and can provide policy implications. First, we considered the compositional and contextual characteristics of area such as mother's demographic, socioeconomic status in the area, fiscal self-reliance of the area, etc. when comparing the DCI between government-supported OUAs and comparison areas, while a series of previous evaluation performed only non-adjusted comparison of DCI ignoring the disadvantageous conditions of OUAs and presented a concern for still low DCI in OUAs. Adjusted comparison of DCI for a range of individual and area-level characteristics would enable achievement of a support program to be evaluated in a fair way. Another strength is that our analyses were based on not sample data but population data, which means there is no concern about generalizability.

Our finding calls our attention to the need for reorienting the goal of the support program. For those areas where there is no maternal hospital, but mothers can easily access the nearest hospital in neighboring cities within a range of time deemed to be safe, ensuring more comfortable travel by providing transportation or subsidizing transportation expenses rather than opening new local hospital would be a more reasonable way to ensure accessibility while better serving their needs. On the other hand, for newly defined OUAs where physical distance is a true barrier to the use of obstetric service, the government needs to expand investment regardless of immediate return on investment so that hospitals are able to provide comprehensive perinatal care that can meet the mother's changing need. Specifically, current funding should be expanded to support hiring additional medical staffs such as pediatrician and anesthesiologist, and also for covering other input costs. This can ensure the steady and sustainable operation of hospitals while gaining trust for the government-supported hospital from mothers. Securing accessibility to adequate maternal services for mothers in underserved areas even though they are small in numbers is what the government is supposed to do.

Further study is needed to explore the break-even point of the maternal hospitals where the hospital considers going out. This would provide information on the minimal amount of investment for government-supported hospital to meet the mothers' demand.

## Conclusion

An increasing number of maternal hospitals in remote areas is going out of business from the ever-declining fertility rate, which causes a concern that mothers in those areas are losing access to basic maternal services. However, our study found evidence that in most current OUAs, physical distance is no longer a barrier to access to maternal service. Mothers' new perception of childbearing and delivery is raising mothers' demands to higher standards of quality and offerings, which resulted in a change of mother's priority in choosing a hospital. On the

other hand, the government-support program was proved to be effective, unlike the government' concern. Our study results suggest that narrow-down of the list, which can capture only true OUAs, and focused investment on them are required to serve those in true need in a sustainable way.

## Supporting information

**S1 File. Original questionnaires in Korean and translated in English.**
(DOCX)

**S1 Table. Characteristics of interviewees.**
(DOCX)

## Author Contributions

**Conceptualization:** Hwa-Young Lee, Nan-Hee Yoon, Juhwan Oh.

**Formal analysis:** Hwa-Young Lee, Nan-Hee Yoon.

**Investigation:** Hwa-Young Lee, Nan-Hee Yoon.

**Methodology:** Juhwan Oh.

**Supervision:** Juhwan Oh.

**Validation:** Juhwan Oh, Joong Shin Park, Jong-Koo Lee, J. Robin Moon, S. V. Subramanian.

**Writing – original draft:** Hwa-Young Lee.

**Writing – review & editing:** Juhwan Oh, Joong Shin Park, Jong-Koo Lee, J. Robin Moon, S. V. Subramanian.

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
