## [Decision Letter · Decision Letter 0]

12 Mar 2020

PONE-D-19-31283

Are “Obstetrically Underserved Areas” really underserved? Role of a government support program in the context of changing landscape of maternal service utilization in South Korea: A sequential mixed method approach

PLOS ONE

Dear Professor Oh,

Thank you for submitting your manuscript to PLOS ONE. After careful consideration, we feel that it has merit but does not fully meet PLOS ONE’s publication criteria as it currently stands. Therefore, we invite you to submit a revised version of the manuscript that addresses ALL points raised during the review process.

We would appreciate receiving your revised manuscript by Apr 26 2020 11:59PM. To enhance the reproducibility of your results, we recommend that if applicable you deposit your laboratory protocols in protocols.io, where a protocol can be assigned its own identifier (DOI) such that it can be cited independently in the future. For instructions see: http://journals.plos.org/plosone/s/submission-guidelines#loc-laboratory-protocols

We look forward to receiving your revised manuscript.

Kind regards,

Frank T. Spradley

Academic Editor

PLOS ONE

Journal Requirements:

2. Please address the following:

- Please ensure you have thoroughly discussed any potential limitations of this study within the Discussion section.

- Please include additional information regarding the interview guide used in the study and ensure that you have provided sufficient details that others could replicate the analyses. For instance, if you developed a questionnaire as part of this study and it is not under a copyright more restrictive than CC-BY, please include a copy, in both the original language and English, as Supporting Information. In addition, please provide any details of pre-testing.

"NO authors have competing interests"

We note that one or more of the authors are employed by a commercial company: Bronx Partners for Health Communities New York City

Reviewers' comments:

Reviewer's Responses to Questions

**Comments to the Author**

1. Is the manuscript technically sound, and do the data support the conclusions?

Reviewer #1: Yes

2. Has the statistical analysis been performed appropriately and rigorously? 

Reviewer #1: Yes

3. Have the authors made all data underlying the findings in their manuscript fully available?

Reviewer #1: Yes

4. Is the manuscript presented in an intelligible fashion and written in standard English?

Reviewer #1: Yes

5. Review Comments to the Author

Reviewer #1: Reviewer’s report: Richard Kalisa MD, PhD

Thank you for the opportunity to review this manuscript

The manuscript provides an in-depth evaluation of the effectiveness of the government support program for obstetrically under served areas in South Korea. It highlights the facilitators and barriers to use of government-supported hospital and the importance of the area-level in explaining the maternal health services utilization in OUAs. The findings are useful to the program and could support its continued implementation and improvement. Overall, the paper is well-written. However, a few issues need consideration:

1. Qualitative arm of the study: The authors should provide details on the sampling of the study participants, rigour/trustworthiness of their qualitative study findings and provide illustrative quotes for supply-side barriers to the success of the program. In addition, under “preference for a specialized maternal hospital over a general hospital” they should label the quote appropriately.

2. In the discussion, it would be important to see a comparison between this study findings and previous evaluations conducted on the program. What are some of the similarities and differences among the different evaluations – findings and recommendations?

3. Results: Revise the sentence “Specifically….” to read better

4. Discussion: The recommendation on “…just making their travel more comfortable by providing transportation itself or subsidizing transportation expenses rather than persuading them to use local hospital in their residential area would be more reasonable way to ensure accessibility while better serving their needs.” might negate the importance of physical/geographical accessibility. Also, could women still utilizing city hospital be related to birth preparedness and complication readiness?

5. The manuscript needs an entire revision of typographical errors

6. Add the study strength and suggestions for further studies

7. The authors should consider using PLoS ONE reference style

6. PLOS authors have the option to publish the peer review history of their article (what does this mean?). If published, this will include your full peer review and any attached files.

Reviewer #1: Yes: Richard Kalisa MD, PhD

---

## [Author Response · Author response to Decision Letter 0]

11 Apr 2020

We uploaded responses to reviewers as a separately file.

---

## [Decision Letter · Decision Letter 1]

22 Apr 2020

Are “Obstetrically Underserved Areas” really underserved? Role of a government support program in the context of changing landscape of maternal service utilization in South Korea: A sequential mixed method approach

PONE-D-19-31283R1

Dear Dr. Oh,

We are pleased to inform you that your manuscript has been judged scientifically suitable for publication and will be formally accepted for publication once it complies with all outstanding technical requirements.

With kind regards,

Frank T. Spradley

Academic Editor

PLOS ONE

Reviewers' comments:

Reviewer's Responses to Questions

**Comments to the Author**

1. If the authors have adequately addressed your comments raised in a previous round of review and you feel that this manuscript is now acceptable for publication, you may indicate that here to bypass the “Comments to the Author” section, enter your conflict of interest statement in the “Confidential to Editor” section, and submit your "Accept" recommendation.

Reviewer #1: All comments have been addressed

2. Is the manuscript technically sound, and do the data support the conclusions?

Reviewer #1: Yes

3. Has the statistical analysis been performed appropriately and rigorously? 

Reviewer #1: Yes

4. Have the authors made all data underlying the findings in their manuscript fully available?

Reviewer #1: Yes

5. Is the manuscript presented in an intelligible fashion and written in standard English?

Reviewer #1: Yes

6. Review Comments to the Author

Reviewer #1: The authors have fully addressed all my shared comments.

7. PLOS authors have the option to publish the peer review history of their article (what does this mean?). If published, this will include your full peer review and any attached files.

Reviewer #1: None

---

## [Editor Report · Acceptance letter]

24 Apr 2020

PONE-D-19-31283R1 

Are “Obstetrically Underserved Areas” really underserved? Role of a government support program in the context of changing landscape of maternal service utilization in South Korea: A sequential mixed method approach 

Dear Dr. Oh:

I am pleased to inform you that your manuscript has been deemed suitable for publication in PLOS ONE. Congratulations! Your manuscript is now with our production department. 

With kind regards,

on behalf of

Dr. Frank T. Spradley 

Academic Editor

PLOS ONE